# Using virtual reality to study preference and restorativeness for streetscape vegetation design

Pooria Baniadam[1]*, Ignacio Requena-Ruiz[1], Jean-Marie Normand[2], Daniel Siret[1], Franck Mars[2]

1 Nantes Université, ENSA Nantes, École Centrale Nantes, CNRS, AAU-CRENAU, UMR 1563, Nantes, France, 2 Nantes Université, École Centrale Nantes, CNRS, LS2N, UMR 6004, Nantes, France

* pooria.baniadam@crenau.archi.fr

## Abstract

The presence of small urban green spaces, such as streetscape vegetation, plays a significant role in the daily exposure to nature for a considerable proportion of urban inhabitants across the globe. This study examines how specific design elements (vegetal and non-vegetal) of small urban green spaces influence human preferences and their alignment with perceived restorativeness dimensions. In each of the 30 trials, participants selected their preferred option and gave reasons for their choice based on the four aspects of the Attention Restoration Theory (fascination, coherence, being away, compatibility). The results demonstrate that the absence of a fence was the most preferred option, irrespective of the fence type. Shorter fences and fences that include greenery were found to be significantly more favored than other types, primarily due to the factor of fascination. Conversely, attributes such as metal and high fences were selected less frequently, with coherence identified as the primary reason for this preference. The most preferred vegetation type was trees, which were selected primarily due to their capacity to evoke fascination. In contrast, bushes and grass, which were less favored, were chosen for their contribution to coherence. A medium level of diversity was preferred over high or low levels when the arrangement of vegetation was not regular. Furthermore, random and regular arrangements were less favored than an intermediate level of arrangement. With regard to the Attention Restoration Theory dimensions, fascination was the primary motive for all options except for the regular arrangement. These findings could assist designers of small urban green spaces in creating more restorative environments. Additionally, the study illustrates the value of employing virtual environments in environmental preference research.

## 1. Introduction

At the present time, approximately 60 percent of the global population resides in urban areas, a proportion that is projected to increase in the coming years [1]. This

**Data availability statement:** All relevant data are within the manuscript and its Supporting Information files.

**Funding:** Initials of the authors who received award: DS, FM, PB Grant numbers awarded: PhD fundings within the CNRS 2021 80Prime program, no grant number. The full name of funder: Centre National de la Recherche Scientifique (CNRS), France URL of funder website: https://miti.cnrs.fr/presentation-prime/ The sponsors or funders DID NOT play any role in the study design, data collection and analysis, decision to publish, or preparation of the manuscript.

**Competing interests:** The authors have declared that no competing interests exist.

has resulted in a reduction in exposure to nature, which numerous studies conducted in recent years have demonstrated to be beneficial for human well-being [2–6]. These studies primarily rely on the Attention Restoration Theory (ART), which posits that exposure to natural environments has a restorative effect on human wellbeing by restoring attentional capacity [7,8]. According to this theory, the capacity for directed attention is finite and can be depleted following prolonged engagement in an attention-demanding task. Some environments, referred to as restorative, have the potential to replenish this attentional resource. Based on the principles of the ART, natural environments are classified as restorative environments that affect humans through the following four aspects: "Fascination" (they attract human attention effortlessly), "Being away" (they offer respite from the demands of daily life), "Extent" (they provide coherence and facilitate immersion), and "Compatibility" (they possess characteristics that align with one's desires). To assess the self-perceived restorative potential of an environment, Hartig et al. [9] introduced the 'Perceived Restorativeness Scale' (PRS) which includes four categories similar to those of the ART. However, "Extent" was replaced by "Coherence," which was interpreted as an element of "Extent" and a reliable predictor of environmental preference [10].

Although the majority of studies examining the benefits of natural environments have historically concentrated on natural settings beyond urban areas, such as forests [4,5,11], mounting evidence suggests that urban nature can also serve as a restorative environment [12–14]. In recent decades, the benefits of nature for urban environments have become increasingly recognized, with a growing emphasis on integrating urban and natural environments. However, this integration has not yet reached a satisfactory level of implementation [15,16], as evidenced by studies indicating that for most city inhabitants, regular, weekly interactions with nature remain uncommon [17].

## 1.1 Background

The dominant focus of research on urban nature is on urban parks, urban forests, and other expansive green spaces situated within or in close proximity to urban areas [13,18–23]. However, the majority of time spent outdoors is dedicated to walking or traversing streets [24], which constitute the largest proportion of urban public spaces, surpassing any type of urban green spaces like parks [25]. Within this context, "small urban green spaces" refers to smaller-scale green areas embedded in the urban fabric. Despite their relatively limited size, these green areas contribute to the wellbeing of urban populations on a daily basis due to their close proximity to residential areas [26].

The term "accessibility" encompasses multiple meanings, sometimes used interchangeably with other terms such as "availability" and "permeability" [19,20] or to describe the comparative opportunities people have to benefit from green spaces [22]. In urban studies, "accessibility" is primarily used as an index of how public green spaces are distributed throughout the city, with distance and proximity as the main factors [27]. This measurement of access to natural environments within urban areas is often based on the distance to large urban parks. Including smaller,

more proximate spaces—such as streetscape vegetation and pocket parks—would significantly increase measured accessibility [28].

In two separate studies conducted in the Netherlands and Portugal, participants were presented with the option of selecting either a large public park or several smaller green spaces [29,30]. The results of both studies indicated that two-thirds of the participants expressed a preference for the latter. A comparable outcome was observed in a study conducted in the United Kingdom, where participants expressed a preference for multiple green spaces of varying dimensions over a single large public park [31]. This preference was attributed to the belief that the larger park would be unable to accommodate the diverse needs and preferences of the community, including those related to biodiversity, wildlife conservation, and proximity.

Furthermore, the growing preference for smaller green spaces, including streetscape vegetation, is likely to be reinforced by contemporary urban design principles. This is because such principles encourage the co-development of natural spaces and buildings for ecological reasons, including climate adaptation and the preservation of biodiversity in urban environments [32].

Studies focusing on larger urban nature, such as parks, investigated a range of elements and their effects on restoration, including the type of vegetation, the presence of water, urban facilities, and lighting [33–36]. A lesser number of studies has examined the impact of individual elements within smaller urban green spaces, such as pocket parks or streetscape vegetation. For instance, Lindal & Hartig [37] discovered that the number of trees and the existence of flowerbeds markedly enhance the probability of a street being perceived as restorative. In a similar study, it was observed that trees exert a greater influence on preference in streetscape vegetation compared to flowers and climbing plants (vertical greenery) [37]. Another contributing factor to the preference for streetscape vegetation is the level of wildness, as suggested by research on different vegetation growth conditions [38].

An important element of small urban green spaces which can affect how they are perceived is the manner in which they are separated from their immediate environment. The term "accessibility" defined above could also be referred to this factor and the boundaries between the green spaces and the people. This can be classified into visual accessibility (related to visibility, openness, and enclosure) and physical accessibility [32]. In the current study, the term "accessibility" is employed to signify possibility of entering an urban green space and refers to presence of fence which is one of the primary urban elements that can impede both physical and visual accessibility.

**1.1.1 Preference and restorativeness.** Prior research has demonstrated a robust correlation between the restorative quality of an environment and individuals' preference for it [39–42]. Additionally, people's general preferences for an environment tend to align with their preferences when seeking a place to relieve stress [43]. Another study suggests that predictors of visual preference in urban nature often indicate the presence of characteristics that contribute to a restorative environment [44]. This correlation may similarly be explained by ART, which emphasizes the role of fascination—a potential shared component between preference and restoration—in creating a restorative environment [45].

It has been demonstrated that the restorative value of a scene serves as the underlying reference criterion for participants' preference judgments [45]. Additional research indicates that when participants select an urban green space, they often choose the same locations, regardless of their differing psychological needs [46]. Furthermore, when individuals are attracted to an environment, they tend to spend more time there, thereby amplifying the restorative benefits [47]. This aligns with the primary objective of incorporating such environments into cities.

**1.1.2 Design elements.** A number of studies have demonstrated that different types of vegetation can exert varying effects on mental health and well-being. For example, one study found that although tree canopies have a positive effect on mental well-being and health, exposure to grass is associated with increased psychological distress [48]. In a different study, which compared panoramic images of trees and grass with the same greenery ratio, it was determined that grass had a more pronounced effect on positive affect [49]. Furthermore, a study comparing urban trees and grasslands revealed that adults reported superior subjective health in areas with a considerable number of trees, in contrast to areas that were predominantly grassland [50].

In regard to the impact of streetscape vegetation, a study has demonstrated the considerable influence of street trees on preference, thereby underscoring their pivotal role in urban planning [36]. However, the study also revealed that grass and vertical greenery had comparatively less pronounced effects. Additionally, studies have indicated that the probability of restoration increases with the number of street trees and the presence of flower beds [25]. Another study exploring the relationship between neighborhood greenness, air pollution exposure, and psychological well-being suggests that street trees contribute more to better mental health than grass [51].

In order to develop a more profound comprehension of the qualities that define green spaces, scholarly inquiries have examined the impact of various factors, including enclosure [52], density [53], naturalness [54], and complexity [55]. Nevertheless, research examining the impact of diverse characteristics of small green spaces and their relationship to mental well-being remains limited.

One feature that has been examined in the literature was biodiversity in urban green spaces, which refers to the diversity of flora and fauna. A study conducted in various cities across Italy demonstrated that higher biodiversity in urban green spaces leads to greater well-being and perceived restorativeness [56]. Similarly, a study on urban green space quality conducted in the UK revealed that biodiversity exerted a more pronounced influence on restorative benefits than site facilities such as benches and lighting [57]. Nevertheless, urban green space design predominantly prioritizes plant life, which is why the present study considers only vegetation diversity rather than overall biodiversity. A previous study found that a high level of vegetation diversity can enhance the restorative effect of small urban green spaces [58].

There is a paucity of studies that have compared the effects of different plant arrangements and diversity within urban green spaces. [59] evaluated preferences in digital landscape models presented on a screen based on three levels of plant arrangements, referred to as coherence, and three levels of plant abundance and diversity, referred to as complexity. The results demonstrated that clustered plants were preferred over formal or scattered organizations, and higher complexity was correlated with higher preference ratings [59]. A second study, also utilizing digital models, revealed that when the scene was characterized by an ordered configuration of elements, greater diversity in tree species resulted in a higher level of preference [60]. However, when the scene was not in order, an inverted U-shaped pattern emerged between diversity and preference, meaning that preference initially increased with diversity but then declined as diversity continued to rise. In a recent study, it was demonstrated that a random arrangement of natural elements in streetscape vegetation increased the perceived restorativeness of the environment compared to a regular arrangement based on geometric patterns [58]. Nevertheless, the absence of a medium level prevented the investigation of the relationship between the variables in question, namely its linearity.

Although the present study concentrates on the subject of vegetation, the design and features of fence as a non-vegetal element is regarded as an important contextual factor that influences vegetative perception. Fences could serve to delineate the boundary between a public and a private space. Furthermore, the presence of a fence, in conjunction with a gate, could serve to indicate whether a green space is accessible at a given time of day. Fences can also be regarded as a form of urban furniture, functioning to delineate the pathways that are open for passage rather than to regulate who is permitted to traverse them [61]. The impact of a fence, contingent on the context, can be either negative (decreasing the benefits derived from green spaces) or positive (increasing the sense of security), influencing the degree to which it is preferred [62]. A recent study demonstrated that the presence of fences on the edge of small urban green spaces can significantly reduce their restorative effect [58]. This result calls for more investigation, particularly regarding the characteristics of fences, such as their height, their material and their integration with the vegetation, which could significantly modulate their influence.

### 1.1.3 Using virtual environments to represent green spaces.

The utilization of Virtual Reality (VR) in environmental research has witnessed a notable surge in recent years [63]. VR has the capacity to simulate realistic environments, thereby enhancing fascination and vividness [64]. The term "virtual nature" is often used to describe natural environments presented in VR [65]. Virtual nature can be presented in a variety of forms, classified according to the level of immersion

and the hardware utilized, which encompasses conventional 2D screens, multi-dimensional projections (e.g., CAVE system), and head-mounted displays (HMDs). Another distinction can be made with regard to the type of graphical representation, which can be either 360° depictions (in the form of videos or images) of real environments or computer-generated 3D environments [66]. One study found that computer-generated environments significantly enhance the sense of presence and positive mood in comparison to equivalent 360° videos [67].

A plethora of studies have sought to clarify the similarities between virtual and real nature, with findings suggesting that both have a comparable impact on human mental wellbeing. For instance, an experiment comparing a view of a lake with an immersive virtual representation of the same environment revealed no significant differences in positive and negative emotions between the two conditions [68]. Similarly, another study found no notable discrepancies in perceived restorative characteristics between a real and a virtual garden [69].

Although virtual environments offer significant potential for creating settings where individual attributes or objects can be altered without changing the rest of the scene, they are not as commonly used in choice experiments as other methods in environmental psychology. Instead, most studies rely on photo-montage methods [38,46,70,71]. A recent study comparing VR with traditional methods for evaluating preferences in neighborhood parks found that VR increases participants' confidence in their preferences [72]. Another study comparing immersive virtual environments with still images suggests that VR offers greater validity and a stronger sense of presence [73].

A number of studies have demonstrated the potential of digital scenes that are based on 3D objects, particularly in terms of facilitating straightforward modifications based on experimental variables. Nevertheless, these virtual environments were frequently utilized for the creation of on-screen presentations rather than for the generation of fully immersive VR experiences. For instance, van Vliet [33] employed computer-generated virtual environments to assess the influence of diverse urban park attributes on user preferences. However, participants lacked the opportunity to experience a sense of presence, as they were exposed to recorded videos of these environments rather than engaging directly with the VR settings [33]. There are other similar studies that created 3D environments in order to generate different alternatives for choice experiments, but instead of providing immersive experiences, they presented only exported images to participants [37,74].

## 1.2 The present study

At present, there is a notable deficit in knowledge concerning the manner by which disparate elements and characteristics of small urban green spaces influence their capacity to facilitate restorative experiences. This study employs immersive virtual environments to address this gap in knowledge by investigating the impact of design choices in streetscape vegetation on the restorative quality of urban settings.

One such understudied element is the physical barrier or fence that separates small green spaces from their surrounding environments. The demarcation of boundaries between green spaces and adjacent areas assumes particular significance when vegetation is distributed along streets. In large parks or urban forests, a fence may be perceived as an integral component of the landscape. However, in small green spaces, a fence serves to delineate the boundary, and its presence can significantly impact the restorativeness of the environment. To explore this aspect, the current study will manipulate the height, material of the fence, and integration of greenery into it to investigate how these attributes influence participants' perceptions of small urban green spaces.

Another aspect of the present study concerns the examination of design features related to vegetation. As previously stated, the influence of diverse vegetation types (e.g., grass and trees) may exhibit varying effects on preference and restorativeness. In human-made green spaces, the number of species may be limited, whereas natural environments typically exhibit a high level of diversity, which may result in greater complexity and varied impacts on perception. Furthermore, the distribution of these natural elements—whether organized and regular, as in human-designed spaces, or more arbitrary and informal, as in natural settings—can influence how the environment is perceived or preferred. This study

investigates these attributes to determine how each may enhance or detract from the preference and restorativeness of streetscape vegetation.

Furthermore, according to ART, a number of factors influence the restorative potential of an environment, including fascination, coherence, being away, and compatibility. Given that the environments in this study were manipulated based on design elements, it is essential to understand how each of these elements relates to the different aspects of ART. Consequently, this study not only assesses preferences for the aforementioned design elements but also investigates the reasons behind these preferences.

Accordingly, the primary objective of this study was to assess preferences for various design elements of streetscape vegetation, including fences, vegetation type, level of diversity and arrangement. The secondary objective was to explore the underlying reasons for these preferences based on the four dimensions of ART: fascination, being away, coherence, and compatibility. To achieve these objectives, we address two research questions: (1) How do specific design elements influence preference and restoration in urban streetscapes? (2) Which ART dimensions most strongly mediate these preferences?

This study makes contributions to both theory and practice. Theoretically, it advances environmental psychology research by employing a novel VR-ART framework that systematically links design elements to restorative experience dimensions – addressing a critical gap in evidence-based streetscape design. Methodologically, it demonstrates how immersive VR enables precise manipulation of vegetation variables while maintaining ecological validity, overcoming limitations of traditional photo-elicitation methods. Practically, the findings provide landscape architects and urban planners with empirically-validated guidelines for creating restorative streetscapes.

## 2. Methods

### 2.1 Research framework

This study systematically evaluates how input variables (fence attributes and vegetation features) influence restorative preferences through controlled 3D virtual environments. The VR platform presented streetscape vegetation sceneries where participants completed choice experiments (30 trials), selecting preferred configurations and justifying choices via ART's four dimensions. Output measures quantified both discrete preferences (selection frequency) and restorative quality (ART dimension distribution). This input-process-output structure isolated design impacts while maintaining ecological validity to assess independent effects. Fig 1 diagrams this framework.

### 2.2 Participants

A total of 57 participants were recruited, with an average age of 30.2 years (SD = 8.05) and a nearly equal gender distribution (28 female, 27 male, and 2 who preferred not to disclose). The sample size was determined based on standard practices in simulated nature studies, where 50% of experiments employ samples below N = 60 [34], ensuring comparability while addressing VR methodology constraints. Approximately a quarter of the participants (n = 14) had never used VR before, while the majority (n = 38) had some prior experience, and only 5 had extensive VR experience. Similarly, approximately 20% of participants (n = 11) had minimal or no experience with video games featuring 3D environments. The majority of participants (n = 36) had prior experience with 3D video games, whereas a smaller number (n = 10) had substantial experience in this domain. The study was approved by the Nantes University Ethics and Scientific Integrity Committee (CEDIS, IRB: IORG0011023, decision #26022024−1).

### 2.3 Materials and environments

Two high-resolution virtual environments were created, modeled after real locations in Nantes. Blender was used for 3D modeling, while Unity was employed to add various options, create tasks, and prepare the environments for VR use. Vegetation was sourced from the Unity Asset Store. Participants wore an HTC Vive Pro VR 2 HMD (2448 × 2448 pixels per

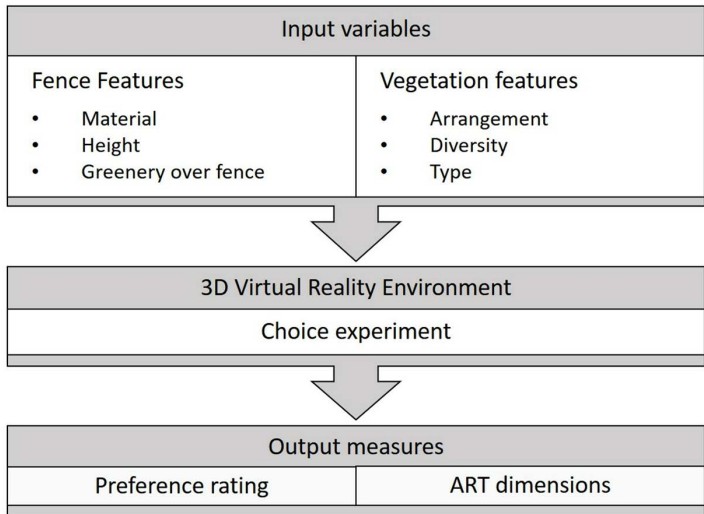

**Fig 1. Research framework diagram.**

eye, 120˚ field of view, 120 Hz refresh rate) to experience the virtual environments. The rendering was conducted using a Dell desktop computer (Core i7 10700 at 2.9 GHz) with an NVIDIA GeForce GTX 3060 graphics card.

The environment utilized for the investigation of factors pertaining to fences was a residential street that features a small vegetated area situated within the concavity of a building (Fig 2). The participants were positioned on the adjacent sidewalk, separated from the green space by a fence (except in the no-fence condition). Three categories of fence attributes were investigated:

1. The material composition of the fences was either metal or wood. The metallic fences were constructed with thin vertical steel bars of a square section linked by rails of a similar character. The wooden fences were made of pickets of equal width (approximately 10 cm) with gaps between them and connected by rails.

2. The height of the structures was divided into two categories: "short" (120 cm) and "high" (210 cm). Each category was constructed with a concrete base measuring 15 cm in thickness at the base.

3. In regard to the subject of vegetation, the fences in question exhibited one of two distinct characteristics. In some instances, vertical greenery was observed to be growing on the fences leading to the designation "with greenery". In contrast, in other instances, the fences displayed a lack of any visible vegetation, leading to the designation "without greenery".

Fence designs replicated common urban configurations from photographic documentation of the study region, with dimensions reflecting standard (though non-regulated) practice to ensure ecological validity while maintaining experimental control.

In the second part of the experiment, which investigated vegetation type, plant diversity and arrangement, an alternative environment was employed—a pedestrian walkway surrounded by three discrete areas of greenery (Fig 3). To investigate vegetation type, three categories of vegetation were utilized: trees, bushes, and grass. The environments utilized for these investigations encompassed two levels of diversity (low and high) and two arrangement settings (regular and random). The "low diversity" level was populated by a single species of each type, whereas the "high diversity" level was populated by six species of trees, seven species of bushes, and three species of grass. In the "regular" arrangement setting, trees and bushes were situated in discrete patches, exhibiting precise alignment and equal distances from one

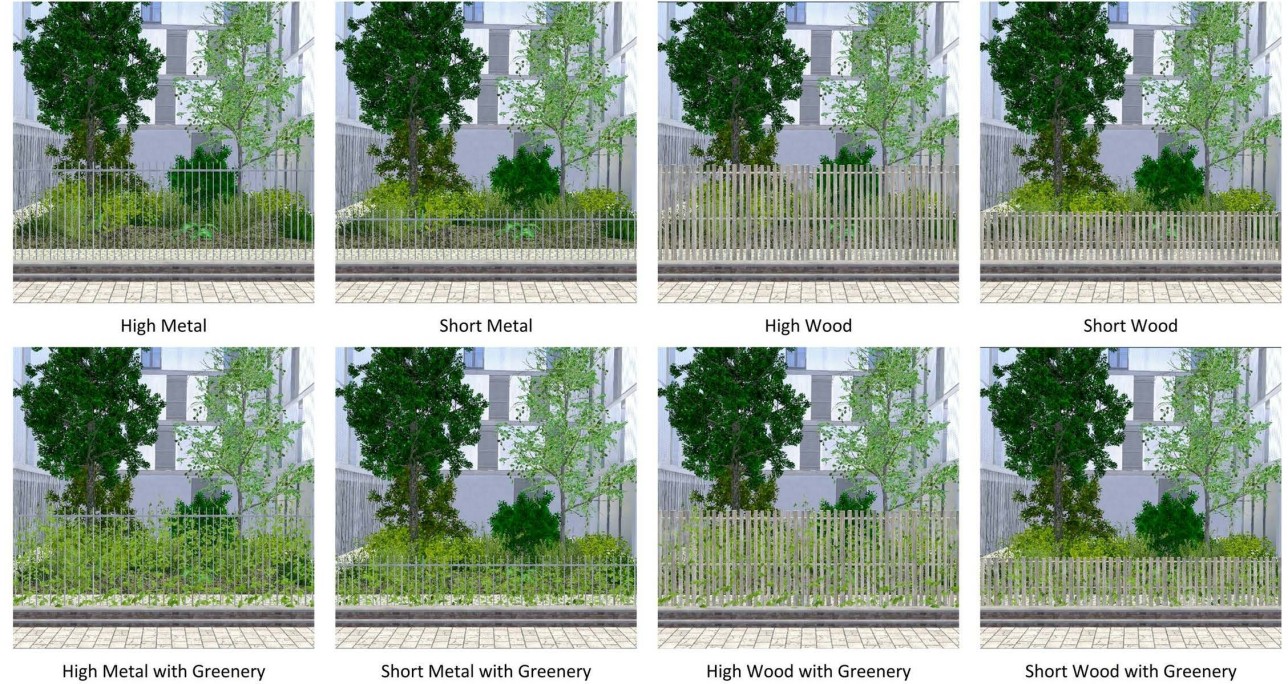

| | | | |
|---|---|---|---|
| High Metal | Short Metal | High Wood | Short Wood |
| High Metal with Greenery | Short Metal with Greenery | High Wood with Greenery | Short Wood with Greenery |

**Fig 2. Eight different conditions of fences based on height, material and containing greenery.**

another. Meanwhile, grass patches were confined to specific, rectangular areas. In contrast, the "random" arrangement involved the arbitrary dispersion of elements across diverse patches, with no restrictions on their positioning or distances between them.

In order to analyze the factors of plant diversity and arrangement, three levels were used for each. Both low and high levels of diversity, and regular and random levels of arrangement were similar to the settings used for investigating vegetation type. The "medium diversity" level contained three species of trees, four species of bushes and two species of grass. The intermediate level of arrangement, "semi-regular", contained trees and bushes integrated in different patches but they were still distributed uniformly in the area. Grass patches were mostly rectangular; however, they were dispersed in different areas. The counts of trees and bushes and area size covering grass were equal in all different levels.

## 2.4 Discrete choice tasks

A discrete choice task is an effective method for identifying whether a subject's preference for one alternative is greater than or comparable to their preference for all other alternatives, as demonstrated by the results of other similar studies [33,37,43,75,76]. By using a 3D environment and a VR HMD, it is possible to change only one or more objects in the environment without changing the rest of the scene or breaking the sense of immersion. The implementation of a controlled experiment of this nature is almost impossible in a real-world setting.

In each task, participants were requested to switch between options in real time while they were immersed in the virtual environment. They were able to move freely inside the virtual environment by physically moving within limits of the experiment room. Participants could view the options from different angles and distances, and choose the one they prefer most. Following the choice of option, participants were asked to indicate one or more of the four statements displayed to them (Fig 4), which were based on four different aspects of the ART and categorized in accordance with the PRS questionnaire

Regular, low diversity Random, high diversity

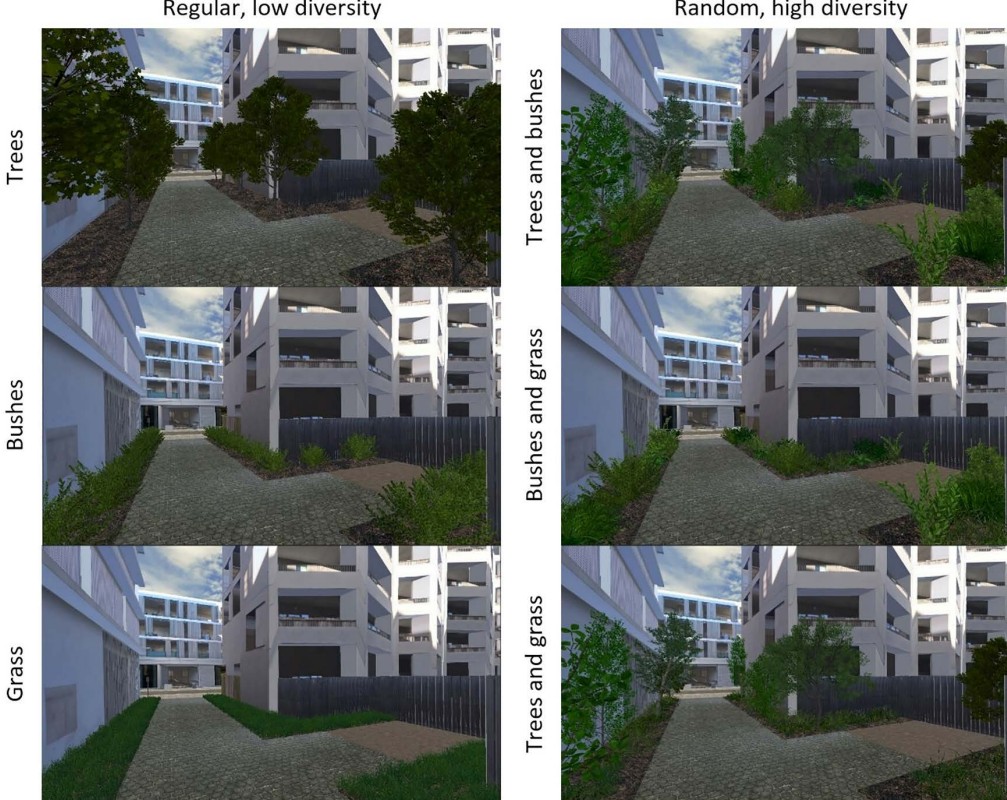

**Fig 3. Example of different options regarding vegetation type.**

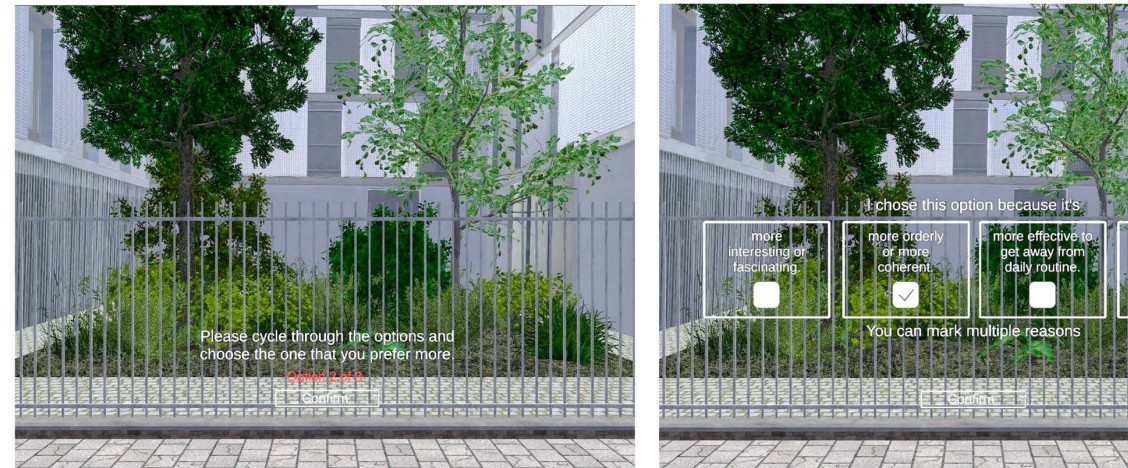

**Fig 4. Choice task experiment and questionnaire presented inside virtual environment.**

[9]. The statements have been formulated to be straightforward and easily comprehensible without necessitating further clarification. The questionnaires were displayed as digital panels within the VR, preventing any disruption to the established sense of immersion. The statements and their associated dimensions are presented in Table 1.

In total, there were 30 distinct tasks based on the aforementioned factors. Tasks 1–16 which pertain to fences are presented in detail in the Table 2, Part 1. These tasks were designed to elicit responses regarding the impact of material, height and presence of greenery on the participant's preference for the fences. The remaining tasks, which focused on vegetation, are presented in Table 2, Part 2. These tasks were conducted to ascertain the influence of vegetation type, diversity, and arrangement on preference.

## 2.5 Procedure

The experiment was conducted on two local university campuses from February 27 to April 12 of 2024. In each of these locations, a room was designated for the experiment, comprising an empty space of at least 2m by 2m to provide sufficient room for the participants to move physically while they were situated within the virtual environments. First, the participants were required to read and sign a written consent form and complete a demographic questionnaire, which inquired about their age, gender, and previous experiences with VR and 3D environments. Subsequently, the participants were provided with contextual guidance regarding the choice tasks. They were instructed to select their preferred option in each task based on the following contextual information: "The environment is part of the route that you traverse on a daily basis and is not your place of residence". The rationale for this guidance was to minimize potential biases related to feeling of safety, maintenance costs, or other factors that were not within the scope of this study.

Subsequently, the participants were instructed to don the VR HMD and engage with a discrete choice task, which was designed to familiarize them with the controller and the questionnaire in the virtual environment. Subsequently, each participant was tasked with completing the entirety of 30 tasks, arranged in a randomized order, which collectively required an average of 15 minutes to accomplish. Participants were permitted to take breaks between tasks and were able to withdraw from the experiment at any point should they encounter any difficulties.

## 3. Results

The results are divided into two sections. The initial section addresses the presence of fences, the various characteristics of fences, and the rationale behind participants' selections related to fences. The subsequent section focuses on vegetation type, plant diversity and arrangement. as well as the motivations behind the choices made for each attribute. In both parts, for each attribute, a binomial test was employed to ascertain whether the most preferred option or the most indicated reason differed significantly from the other options or reasons in the related tasks.

## 3.1 Fence

**3.1.1 Presence of fence.** In tasks 1–8, a comparison was made between two different types of fences and a third option, which presented no fence at all. In all eight tasks, the "no fence" option was selected with a significantly higher frequency than the other two options, with selection rates varying from 63% to 84% (Fig 5).

Table 1. Statements for reasons behind each choice based on the ART dimensions.

| Dimension | The statement presented to the participant: "I chose this option because it is…" |
|---|---|
| Fascination | "…more interesting or fascinating." |
| Coherence | "…more orderly or more coherent." |
| Being away | "…more effective to get away from daily routine." |
| Compatibility | "…more suited to my personality." |

**Table 2. Tasks employed in the experiment along with the options presented in each task.**

Part 1 of the tasks related to fence attributes.

| Subject | Task | Factor under investigation | Option 1 | | | Option 2 | | | Option 3 |
|---|---|---|---|---|---|---|---|---|---|
| | | | Height | Material | Greenery | Height | Material | Greenery | |
| Fence | Task 1 | Height | High | Metal | No | Short | Metal | No | No Fence |
| | Task 2 | | High | Wood | No | Short | Wood | No | No Fence |
| | Task 3 | Material | Short | Metal | No | Short | Wood | No | No Fence |
| | Task 4 | | Short | Metal | No | Short | Wood | No | No Fence |
| | Task 5 | Greenery | High | Metal | No | High | Metal | Yes | No Fence |
| | Task 6 | | Short | Metal | No | Short | Metal | Yes | No Fence |
| | Task 7 | | High | Wood | No | High | Wood | Yes | No Fence |
| | Task 8 | | Short | Wood | No | Short | Wood | Yes | No Fence |
| | Task 9 | Height | High | Metal | No | Short | Metal | No | – |
| | Task 10 | | High | Wood | No | Short | Wood | No | – |
| | Task 11 | Material | High | Metal | No | High | Wood | No | – |
| | Task 12 | | Short | Metal | No | Short | Wood | No | – |
| | Task 13 | Greenery | High | Metal | No | High | Metal | Yes | – |
| | Task 14 | | Short | Metal | No | Short | Metal | Yes | – |
| | Task 15 | | High | Wood | No | High | Wood | Yes | – |
| | Task 16 | | Short | Wood | No | Short | Wood | Yes | – |

Part 2 of the tasks related to vegetation type, diversity and arrangement.

| Subject | Task | Fixed factors | | Option 1 | Option 2 | Option 3 |
|---|---|---|---|---|---|---|
| | | Arrangement | Diversity | | | |
| Vegetation type | Task 17 | Regular | Low | Trees | Bushes | Grass |
| | Task 18 | Random | Low | Trees | Bushes | Grass |
| | Task 19 | Regular | High | Trees | Bushes | Grass |
| | Task 20 | Random | High | Trees | Bushes | Grass |
| | Task 21 | Regular | Low | Trees and bushes | Bushes and grass | Trees and grass |
| | Task 22 | Random | Low | Trees and bushes | Bushes and grass | Trees and grass |
| | Task 23 | Regular | High | Trees and bushes | Bushes and grass | Trees and grass |
| | Task 24 | Random | High | Trees and bushes | Bushes and grass | Trees and grass |
| Diversity | Task 25 | Regular | – | Low diversity | Medium diversity | High diversity |
| | Task 26 | Semi-regular | – | Low diversity | Medium diversity | High diversity |
| | Task 27 | Random | – | Low diversity | Medium diversity | High diversity |
| Arrangement | Task 28 | – | Low | Regular | Semi-regular | Random |
| | Task 29 | – | Medium | Regular | Semi-regular | Random |
| | Task 30 | – | High | Regular | Semi-regular | Random |

**3.1.2 Different types of fences.** The eight distinct types of fences were once more compared in pairs through the completion of tasks 9–16. The results are presented in Fig 6.

In the initial two tasks of this section, the distinction between the two presented options was based on the height. The fences were of analogous material in each task, comprising metal in task 9 and wood in task 10. In both tasks, a majority of the participants (over 80%) indicated a preference for the shorter fence over the taller one. The difference in preference between the two options was statistically significant for both metal (84.2%, 95% CI [0.72, 0.93], $p < 0.001$) and wood (87.7%, 95% CI [0.76, 0.95], $p < 0.001$).

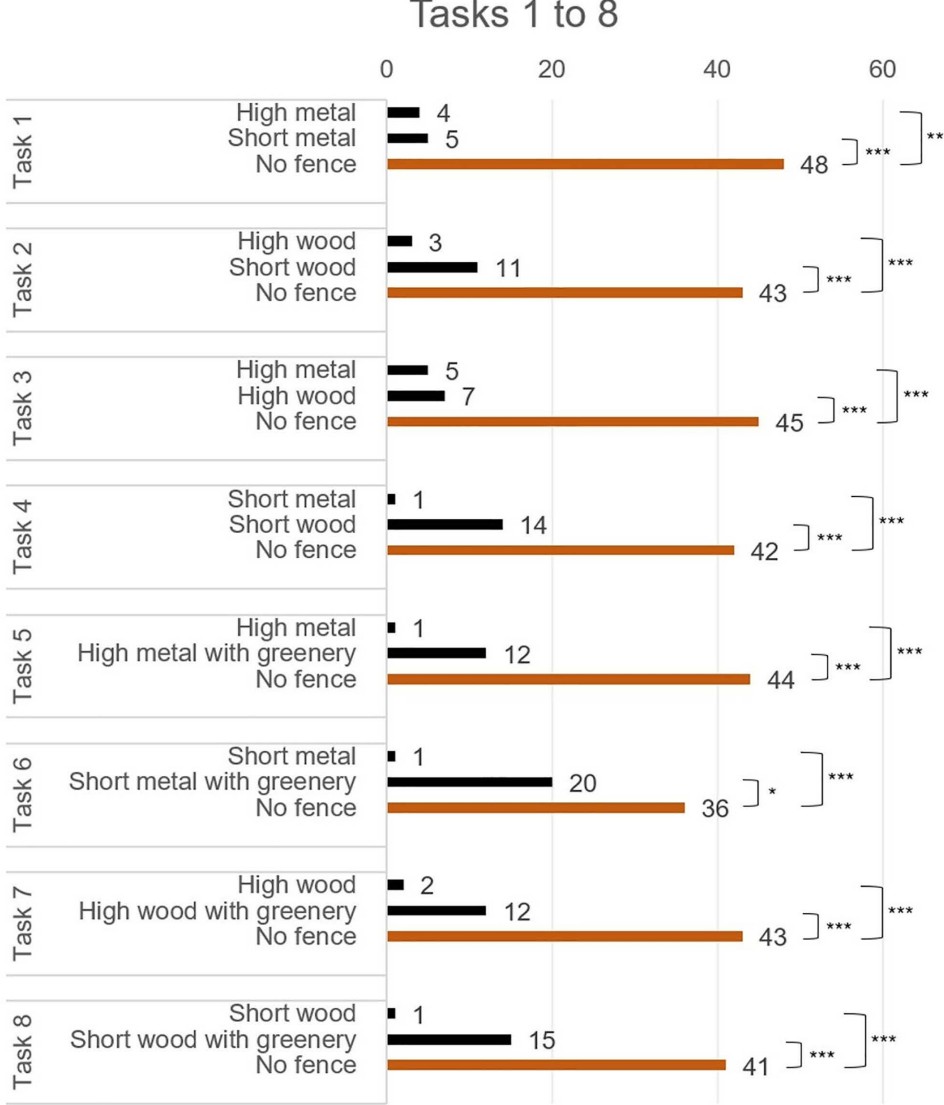

**Fig 5. Results of the tasks 1 to 8 in which absence of a fence were compared with different fence attributes.** Count of the choices are presented along with paired comparisons between options. * p < .05, *** p < .001.

In tasks 11 and 12, participants were presented with the option of selecting either a metal or wooden fence. The height of the fences was held constant across both options, with a taller fence being presented in task 11 and a shorter fence in task 12. When the fence was of a greater height, the difference in preference was not statistically significant, with 49.1% of the participants selecting wood and 50.1% selecting metal. Conversely, when the fence was of a shorter height, the participants demonstrated a significantly higher preference for wood over metal (70.2%, 95% CI [0.57, 0.82], p < 0.003).

In the final four tasks of this section (tasks 13–16), participants were asked to choose between a fence that contained greenery and an identical fence without greenery over it. The tasks were distinguished by the materials and heights of the fences utilized, yet these characteristics remained consistent between the two options within each task. For example, in task 14, a short metal fence was compared to a short metal fence with greenery. The preference for the fence with

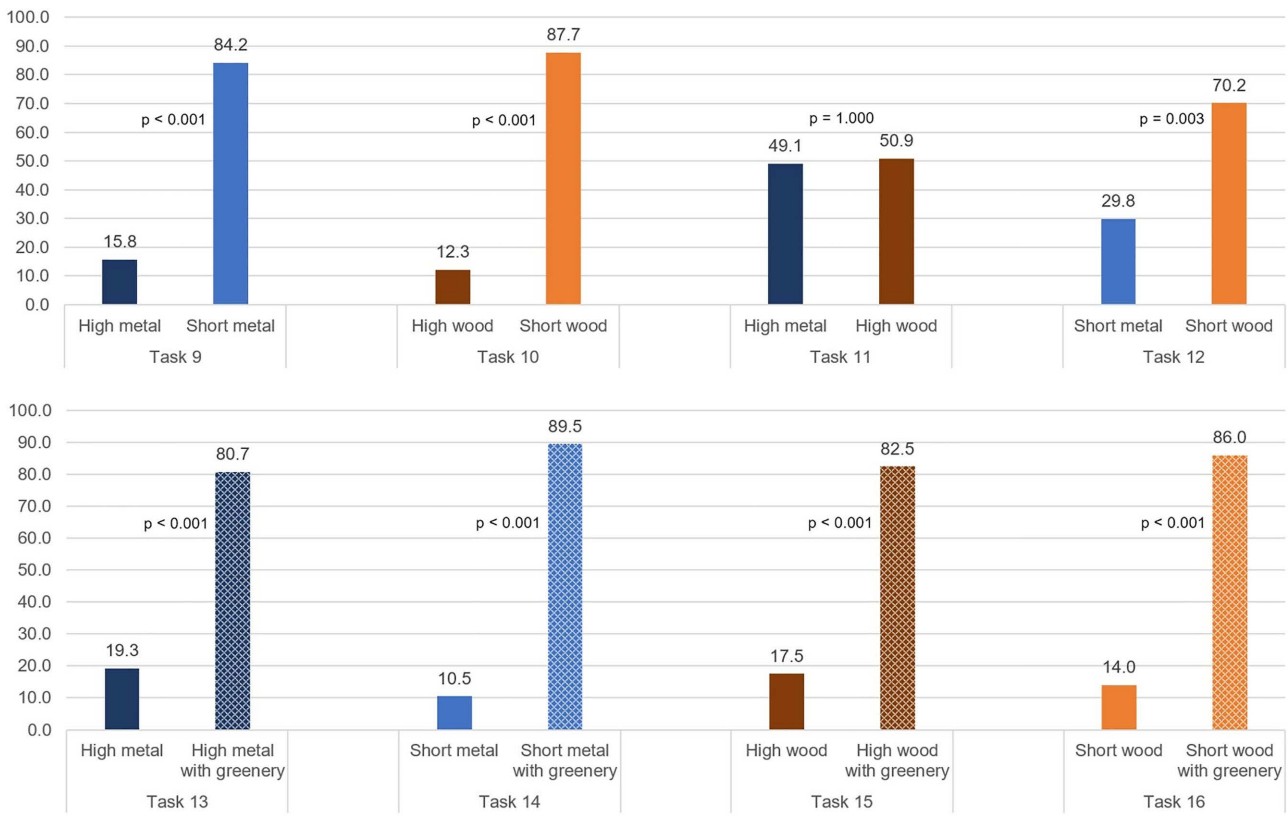

**Fig 6. Results of tasks 9 to 16, in which different fence attributes are compared.** Percentage of the choices are presented along with p-value for each task.

greenery was significant across all four tasks, regardless of the fence type. The preference rates ranged from 80.7% to 89.5%, as illustrated in Fig 6.

**3.1.3 Reasons behind choices for the fence attributes.** A total of seven attributes were evaluated, comprising two based on material, two based on height, two based on greenery, and one regarding absence of a fence (Fig 7). The reasons indicated for choosing absence of a fence was calculated based on the first eight tasks for which the no fence option was presented. For the remaining three attributes, the total number of reasons was calculated based on the categories of each attribute, both when a "no fence" option was or was not included among the available options (Table 3). To take the metal attribute as an example, the sum of the reasons was based on the four tasks in which metal was compared to wood. These were tasks 3, 4, 11 and 12. Although metal fences were also present in other tasks, the reasons indicated for a choice in those tasks were not counted for the metal attribute. This was due to the reason that the comparison in those tasks was between two metal fences, for example, short metal versus high metal, which means that the choice and the reason behind it was not based on the metal attribute.

The results of the binomial tests and the level of significance are presented in Table 3. When participants selected the absence of the fence, they indicated *fascination* and *compatibility* as equally important reasons (64.3%), which were significantly more frequently stated than *being away* ($\chi^2(1) = 6.69$, $p < .01$) and even more frequently than *coherence* ($\chi^2(1) = 46.81$, $p < .001$).

The primary motivation for selecting metal fences was *coherence*, with a rate of 72.6%, which was significantly higher than the rates for the remaining three reasons. However, when participants expressed a preference for wood, they

|  | Fascination | Coherence | Being Away | Compatibility |
|---|---|---|---|---|
| No Fence | 64.3 | 28.7 | 49.4 | 64.3 |
| Metal | 27.5 | 72.5 | 11.8 | 29.4 |
| Wood | 53.3 | 25.6 | 18.9 | 58.9 |
| Short | 51.8 | 46.5 | 26.3 | 50.9 |
| High | 26.1 | 91.3 | 4.3 | 4.3 |
| With greenery | 78.2 | 25.8 | 36.9 | 50.8 |
| Without green | 30.0 | 67.5 | 7.5 | 45.0 |
| Trees | 68.8 | 17.4 | 48.6 | 54.3 |
| Bushes | 31.9 | 56.9 | 30.6 | 45.8 |
| Grass | 38.9 | 50.0 | 16.7 | 44.4 |
| Trees and Bushes | 79.3 | 27.4 | 42.2 | 57.8 |
| Bushes and Grass | 40.0 | 43.3 | 23.3 | 76.7 |
| Trees and Grass | 52.4 | 61.9 | 31.7 | 49.2 |
| Low diversity | 46.3 | 31.7 | 34.1 | 14.6 |
| Medium diversity | 80.0 | 70.7 | 45.3 | 48.0 |
| High diversity | 45.5 | 41.8 | 20.0 | 36.4 |
| Regular | 44.8 | 65.5 | 20.7 | 48.3 |
| Semi-regular | 67.4 | 40.4 | 38.2 | 59.6 |
| Random | 64.2 | 35.8 | 32.1 | 52.8 |

**Fig 7. Reasons indicated by the participants for their choices for the design elements.**

indicated that *compatibility* was the primary reason (58.9%), which was significantly higher than *coherence* ($\chi^2(1)$ = 11.84, p < .001) and *being away* ($\chi^2(1)$ = 18.51, p < .001). The second most frequently cited reason was *fascination* (53.33%), which was not significantly lower than *compatibility* ($\chi^2(1)$ = 0.25, p = 0.619).

**Table 3. Main reasons indicated for each fence attributes compared to other reasons.**

| | Tasks considered | Highest stated reason | Rate | χ² paired with other reasons | | | |
|---|---|---|---|---|---|---|---|
| | | | | Fascination | Coherence | Being Away | Compatibility |
| No fence | 1–8 | Fascination Compatibility | 64.33 | – | 46.81*** | 6.69* | – |
| Metal | 3, 4, 11, 12 | Coherence | 72.55 | 10.37*** | – | 22.35*** | 9.31** |
| Wood | 3, 4, 11, 12 | Compatibility | 58.89 | 0.25 | 11.84*** | 18.51*** | – |
| Short | 1, 2, 9, 10 | Fascination | 51.75 | – | 0.32 | 9.45** | 0.009 |
| High | 1, 2, 9, 10 | Coherence | 91.30 | 8.33** | – | 18.18*** | 18.18*** |
| With greenery | 5 to 8, 13–16 | Fascination | 78.17 | – | 66.50*** | 37.30*** | 14.65*** |
| Without greenery | 5 to 8, 13–16 | Coherence | 67.50 | 5.77* | – | 19.2*** | 1.80 |

Note. *p < .05, **p < .01, ***p < .001.

When participants indicated a preference for a short fence, the primary motivation behind this preference was *fascination* (51.8%), although this was not indicated significantly more often than other reasons, with the exception of *being away*, which was the least indicated reason ($\chi^2(1) = 9.45$, p < 0.01). The participants who selected a high fence indicated, by a significant margin, that their primary reason for this choice was *coherence*. This was given significantly more often than the other three reasons, similar to the pattern observed for the preference for a metal fence.

With regard to the attribute indicating the presence of greenery over the fence, the majority of participants (78.2%) indicated that they were driven by a sense of *fascination*, which was significantly more prevalent than the other stated reasons. Finally, when participants selected an attribute devoid of greenery, the primary rationale provided was *coherence* (67.5%), which was stated significantly more frequently than *fascination* ($\chi^2(1) = 5.77$, p < 0.001). 5) and *being away* ($\chi^2(1) = 19.2$, p < 0.001), but not *compatibility* ($\chi^2(1) = 1.8$, p = 0.18).

### 3.2 Vegetation

**3.2.1 Vegetation type.** The initial four tasks in this section (tasks 17–20) undertake a comparative analysis of three distinct types of vegetation (trees, bushes, and grass) across four distinct scene characteristics, with a consideration of both arrangement and diversity (Fig 8A). In all four tasks, the highest rating was assigned to "trees", with a range of 53% to 67%. According to the results of the paired comparisons with other types (Table 4), the difference was statistically significant with the exception of the *random – high diversity* scene. In the *regular – low diversity* scene, "grass" was selected as the second preferred option (25%), though it was not significantly higher than "bushes" (14%). In the other three scenes, participants selected "grass" significantly less often than the other two types, indicating that "grass" was retained as the least preferred vegetation type (rates varying from 0% to 5%).

In the subsequent four tasks, a comparable evaluation was conducted, with each option comprising two distinct vegetation types (Fig 8B). In three of the scenes, the highest-rated option was "trees and bushes" (65% to 72%), which was selected significantly more frequently than the other two options. In *regular – low diversity* which was the single exception where the most preferred option was "trees and grass" (51%), this was not chosen significantly more often than "trees and bushes", which came second (33.3%). The least preferred option in this scene was "bushes and grass" (16%), which was selected significantly less frequently than "trees and grass" but not "trees and bushes". In both the *random – low diversity* and *regular – high diversity* scenes, the least favored option was "bushes and grass" (9% and 11% respectively). However, it was selected with a frequency not significantly less than the second option, "trees and grass". Lastly, in the *random – high diversity* scene, the second and third preferred options were "bushes and grass" (18%) and "trees and grass" (16%). Nevertheless, the difference between these two options was not statistically significant (Table 4).

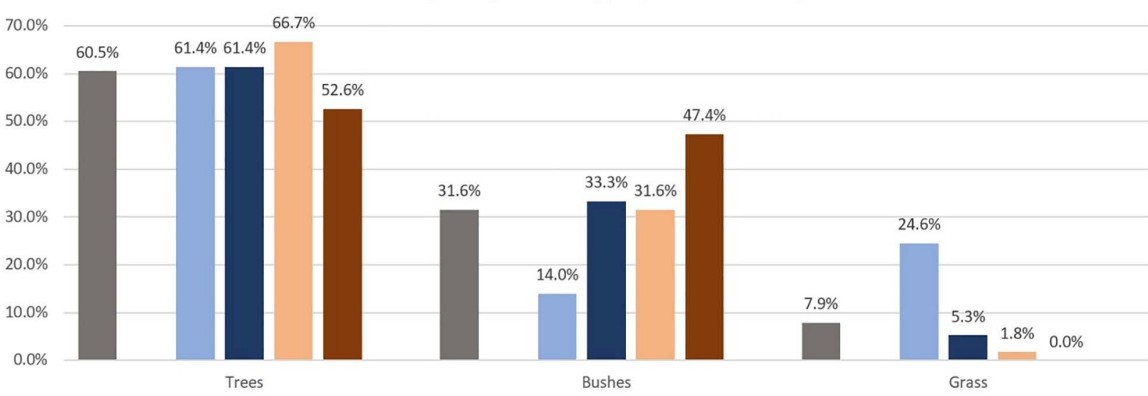

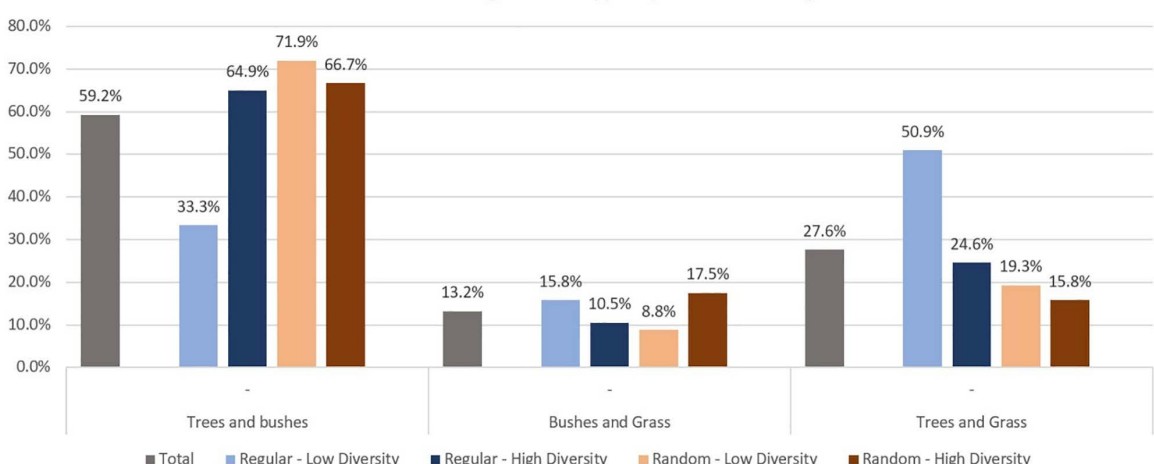

**Fig 8. Results of tasks 17 to 24, in which different types of vegetation are compared based on different conditions of diversity and arrangement.**

**Table 4. Paired binomials between two options about vegetation types.**

| | Regular, low diversity | | Random, low diversity | | Regular, high diversity | | Random, high diversity | |
|---|---|---|---|---|---|---|---|---|
| | $x^2$ | p | $x^2$ | p | $x^2$ | p | $x^2$ | p |
| Trees vs bushes | 16.95 | <.001 | 7.14 | <.01 | 4.74 | <.05 | 0.16 | 0.69 |
| Trees vs grass | 9.00 | <.01 | 35.10 | <.001 | 26.95 | <.001 | 30.00 | <.001 |
| Bushes vs grass | 1.64 | 0.20 | 15.21 | <.001 | 11.63 | <.001 | 27.00 | <.001 |
| TB vs TG | 2.08 | 0.15 | 17.31 | <.001 | 10.37 | <.01 | 17.89 | <.001 |
| TB vs BG | 3.57 | 0.06 | 28.17 | <.001 | 22.35 | <.001 | 16.33 | <.001 |
| TG vs BG | 10.53 | <.01 | 2.25 | 0.13 | 3.20 | 0.07 | 0.05 | 0.82 |

TB = trees and bushes, TG = trees and grass, BG = bushes and grass.

**3.2.2 Reasons behind choices for vegetation type.** Based on the sum of the reasons stated for preferring each vegetation type in tasks 17–20 (Fig 7), the most and least frequently indicated reason for "trees" was *fascination* (68.8%), which was marked significantly more often than *coherence* ($\chi^2(1) = 42.36$, p < .001) and *being away* ($\chi^2(1) = 4.84$, p < .05). However, the second most frequently indicated reason for selecting "trees", which was *compatibility* (54.3%), was not significantly less frequent than *fascination*. The least frequently selected reason for preferring "trees" was *coherence* (17.4%), which was significantly less prevalent than the other three reasons.

The primary motivation for participants to prefer "bushes" was *coherence*, which was selected 56.9% of the time. This was not significantly more prevalent than the second-most common motivation, *compatibility* (45.8%), but was significantly higher than the other two reasons, *fascination* ($\chi^2(1) = 5.06$, p < .05) and *being away* ($\chi^2(1) = 5.73$, p < .05). No significant differences were observed between the remaining reasons. With regard to "grass", the total number of stated reasons was relatively low, as the highest-rated reason, *coherence*, was given by only nine respondents. Consequently, no significant differences were identified between the motives stated for "grass".

Similarly, the motivation behind the selection of mixed vegetation types in tasks 21–24 (Fig 7) was examined, and the discrepancies between the justifications for each option were quantified. The primary motivation for selecting the "trees and bushes" option was *fascination*, as indicated by 79.3% of participants, which was significantly more prevalent than the other three reasons. This was also identified as the highest rate for any stated reason regarding vegetation type. In contrast, *coherence* was the least frequently considered motive, indicated by only 27% of participants and significantly less than the other stated reasons.

The highest marked reason for "trees and grass" was coherence (61.9%), whereas for "bushes and grass," it was compatibility (76.7%). For both options, the least frequently selected reason was being away (31.7% and 23.3%, respectively). The only significant difference in the paired reasons was between the highest and lowest marked reasons ($\chi^2(1) = 6.12$, p < .05; $\chi^2(1) = 8.53$, p < .01).

**3.2.3 Plant diversity and arrangement.** The final six tasks (tasks 25–30) were designed to ascertain preferences regarding diversity and arrangement. Tasks 25–27 each presented three options regarding three levels of diversity, which were to be selected by participants based on their preferences. The three tasks differed from one another based on the level of arrangement (Fig 9A).

The results of the multinomial tests indicated that only task 27, which was based on a random arrangement, exhibited a distribution of responses that was significantly disparate from an equal distribution ($\chi^2(1) = 11.47$, p = 0.003). In this task, the most preferred option was medium diversity (54.4%), which was selected significantly more often than both low diversity ($\chi^2(1) = 8.40$, p < .01) and high diversity ($\chi^2(1) = 6.42$, p < .05). For tasks 25 (regular) and 26 (semi-regular), no statistically significant differences were identified between the selected options ($\chi^2(1) = 2.21$, p = 0.331). Although not statistically significant, the medium diversity level was rated higher than the other two levels in task 26, a similar pattern that was observed in task 27. In all three tasks, the least preferred option was "low diversity".

In the subsequent and concluding three tasks, participants were presented with options comprising varying degrees of arrangement, designated as "regular," "semi-regular," and "random." The three tasks differed based on three levels of diversity (Fig 9B).

In all three tasks, the "semi-regular" arrangement was the most preferred option, with rates varying from 45.6% to 64.9%. In Task 28 (*low diversity*), the rates for the "regular" and "random" arrangements were found to be equal, which was significantly lower than that observed for the "semi-regular" arrangement ($\chi^2(1) = 15.51$, p < .001). In Task 29 (*medium diversity*), the least preferred option was "regular" (22.8%), which was selected at a significantly lower rate than "semi-regular" ($\chi^2(1) = 4.33$, p < .05). Similarly, in Task 30 (*high diversity*), the least preferred option was "regular" (10.5%), which was significantly less preferred than both "semi-regular" ($\chi^2(1) = 12.50$, p < .001) and "random" ($\chi^2(1) = 11.64$, p < .001).

**3.2.4 Reasons behind the choices for diversity and arrangement.** The reasons for three different levels of diversity were accumulated through tasks 25–27, where these levels were compared to each other (Fig 7). In all three levels,

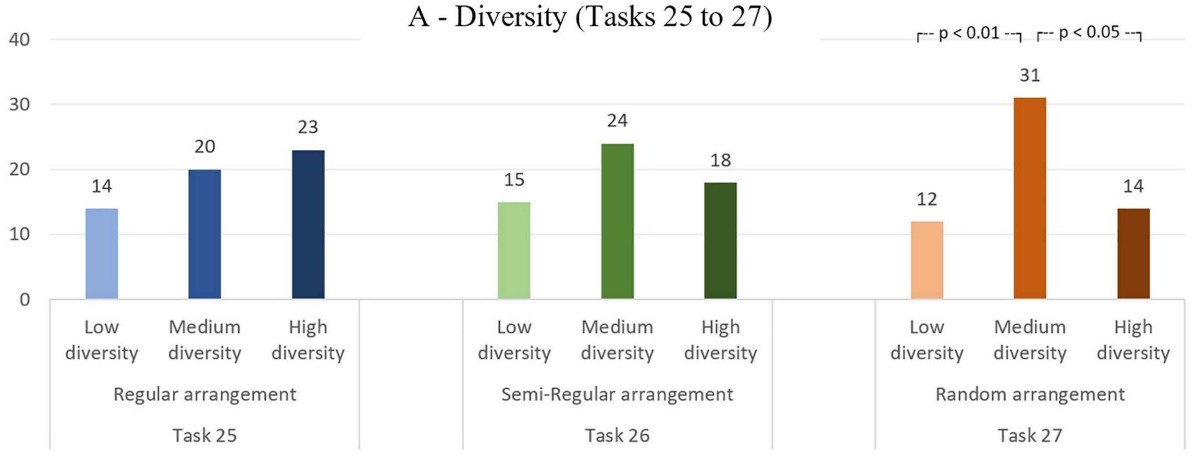

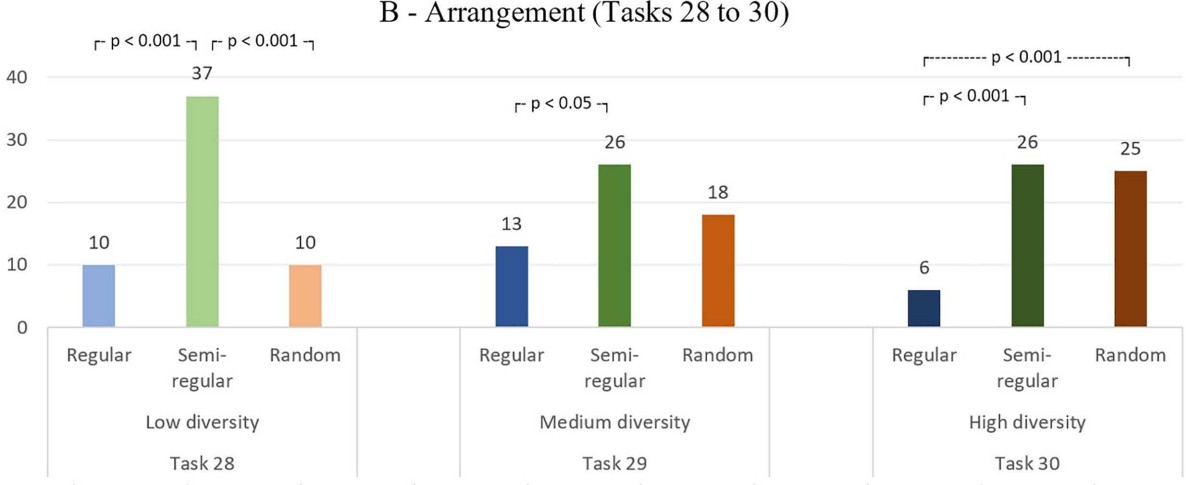

**Fig 9. Results of the tasks 25 to 30 in which different conditions of diversity and arrangement are compared.**

highest motive was *fascination*. For "low diversity", *fascination* was chosen significantly more often than the remaining three reasons: *being away* ($\chi^2(1) = 5.06$, $p < .05$), *coherence* ($\chi^2(1) = 10.97$, $p < .001$), and *compatibility* ($\chi^2(1) = 20.48$, $p < .001$). The least frequently stated reason, which was *compatibility*, was also significantly less prevalent than the reason of *being away* ($\chi^2(1) = 6.13$, $p < .05$).

For "medium diversity", both the first and second highest stated reasons, *fascination* (80.0%) and *coherence* (70.7%), were significantly higher than the two other reasons, *compatibility* (48.0%) and *being away* (45.3%). This pattern was repeated for "high diversity", although rates were lower. The highest and lowest rated reasons, *fascination* and *being away*, were indicated, respectively, by 45.5% and 20.0% of the participants who preferred this level of diversity.

A similar analysis demonstrated which motivations were behind the preference for the three levels of arrangement that were presented as options in tasks 28–30. For the "regular" arrangement, the highest stated reason was *coherence* (65.5%), though it was marked only significantly more often than the least stated reason, *being away* (38.2%; $\chi^2(1) = 6.76$, $p < .01$). In both the "semi-regular" and "random" arrangements, the highest-rated motive was *fascination* (67.4% and 64.2%, respectively). This reason was selected significantly more often than the two least-rated motives, *coherence* and *being away*.

## 4. Discussion

### 4.1 Fences

One of the objectives of this study was to gain a deeper comprehension of the preferences associated with the fencing of small green spaces in urban contexts. The initial finding indicated a proclivity for the absence of fencing, suggesting that, irrespective of the material, height, or presence of vegetation, individuals exhibited a preference for an unobstructed connection with green spaces as they encountered them in urban environments. This outcome can be attributed to the influence of accessibility, a factor that has been previously identified in research [32].

The results indicated a significant preference for short fences over high ones, regardless of the material, when a fence was present. This finding is consistent with the overall preference for the absence of any fence, as participants indicated a preference for minimal physical or visual barriers between themselves and the greenery. The analysis additionally indicated that, in the case of short fences, there was a notable preference for wooden structures over their metal counterparts. However, when the fence was of a considerable height, no material preference was observed. The preference for wood over metal was anticipated, as wood is more natural character and fosters greater coherence with green elements. The absence of this effect for high fences might be attributed to the thicker components of wooden fences in comparison to metal ones, which significantly reduced visual accessibility. Moreover, metal fences may appear more congruent with the surrounding architecture, which is often characterized by cold grey tones (metal and concrete), making them appear more typical in such contexts. This hypothesis is supported by the fact that the participants who preferred metal indicated coherence as the main motivation for their choice.

The addition of greenery to fences, irrespective of their height or material, was consistently perceived as a positive intervention by participants, who expressed a preference for vegetated fences over their non-vegetated counterparts. Although there is a lack of empirical studies that have specifically examined the effect of greenery on fences in terms of preference and restorativeness, research on similar elements, such as green façades or green walls, has demonstrated comparable positive effects when compared to non-vegetated structures [77,78].

An analysis of the reasons provided by participants to justify their choices indicated that the no fence condition was selected primarily due to its perceived fascination, compatibility with their personality, and, to a lesser extent, its ability to provide a sense of being away from ordinary life. Only a few participants selected the absence of a fence due to its coherence. Conversely, fences that were not generally well-received, such as high or metal fences, were chosen by some for their coherence. This indicates that individuals who seek fascination and compatibility tend to prefer the absence of physical barriers, whereas those who seek coherence tend to prefer a functional fence, namely a physical barrier that is more robust and effectively prevents access to a protected green space.

Although material was identified as the least significant factor in fence preferences, the underlying reasons for selecting different materials varied. Individuals who expressed a preference for wood over metal indicated that *fascination* and *compatibility* were the primary reasons for their choice, which was similar to the preferences expressed for the no-fence condition. In contrast, those who selected metal fences did so primarily due to *coherence*, as previously mentioned. It is noteworthy that the motive of *being away* was the least frequently cited reason for selecting either wood or metal. This pattern was also observed with regard to preferences on the height of the fence. It can be concluded that the presence of fences, regardless of their material or height, is inherently in opposition to the feeling of escape associated with the concept of *being away*. The absence of a fence was the primary condition that fostered this sense, as approximately half of the participants who selected the "no fence" option indicated this reason. To a lesser extent, fences with greenery also supported the *being away* experience, with one-third of participants who selected this option citing this motive. However, the primary reason for preferring fences with greenery was *fascination*, cited by 78% of participants, which was a higher rate than the rate of *fascination* for the no-fence condition (64%).

## 4.2 Vegetation type

In regard to vegetation types, the majority of participants expressed a clear preference for trees, regardless of the diversity or arrangement of the vegetation in the scene. The importance of trees for both preference and restorativeness has been highlighted in numerous studies [25,33,76,79]. However, these findings are at odds with those of other research projects that have compared different types of vegetation. For example, in a VR experiment, Huang et al. [49] found that grass was more restorative than trees. Another study demonstrated that when individuals sought restoration in small urban parks, grass was identified as the most important element, even surpassing trees, whereas bushes were considered the least significant [75]. It is important to note that in the former study, grass was depicted as expansive, uniform areas, whereas in the latter, participants were provided only with written descriptions of the vegetation, devoid of any accompanying visuals. This suggests that factors such as the manner in which grass is visually represented, as well as scale and context, could play a pivotal role in the comparison of vegetation types. This discrepancy was also observed in the present study, wherein the preferences between grass and bushes were significantly influenced by the diversity and arrangement of the vegetation. In the regular–high diversity condition, grass was selected by a quarter of the participants, whereas in the other conditions, the rate of preference for this vegetation type was almost zero. It can be inferred that grass is perceived as favorable when it is evenly distributed in the environment and does not contain other types of low-level vegetation, such as weeds, which would be present when the level of diversity is higher. This observation could be further investigated by examining the aspects of wildness and the duality of wild vs. tended nature in urban environments, as higher diversity may be perceived as wild and subsequently as not clean or safe [80].

## 4.3 Plant diversity and arrangement

In the absence of a regular arrangement of vegetation, a mid-range level of diversity was preferred over low or high diversity. This finding aligns with the results of another study, which demonstrated that higher diversity was preferred only when the scene was highly ordered; otherwise, as diversity increased, the preference initially rose but eventually declined [60]. Additionally, the present study demonstrated that a semi-regular arrangement was preferred over a random or regular configuration in all diversity conditions. Prior investigations have indicated that exceedingly high or low levels of complexity may result in diminished aesthetic appreciation [81]. This can be in line with the results of the present study as diversity and arrangement could be assumed as important factors of complexity in a green space. Previous studies have identified a similar inverted-U-shaped pattern in preferences with regard to green space features such as complexity, density, and openness [82–85].

## 4.4 Theoretical and practical implications

From a design perspective, the results offer practical guidance for urban planners and landscape architects. When designing small urban green spaces, it is advisable to avoid extremes: very limited diversity paired with rigid, regular arrangements may create sterile, uninspiring spaces, while overly diverse and chaotic arrangements may appear unkempt or overwhelming. Instead, incorporating a moderate number of plant species arranged in a semi-random pattern can achieve the dual goals of naturalness and order, enhancing both the aesthetic and restorative qualities of the space.

These ART-informed findings translate to actionable design strategies for supporting population mental well-being. Small urban green spaces with moderate level of diversity could create 'micro-restorative opportunities' during daily activities like commuting. Our results suggest arranging these elements at a medium regularity along pedestrian corridors would maximize their restorative value. While the present study examined immediate preferences, the design principles emerging from the VR environment could mirror features of real-world green spaces associated with longitudinal improvements in well-being, as evidenced by studies [63,86]. It is imperative that future research directly examine the performance of these specific configurations in longitudinal, real-world settings.

The findings of this study indicate that it would be more prudent for designers to refrain from incorporating fences within newly developed small urban green spaces. In certain circumstances, the construction of barriers between green spaces and adjacent areas may appear to be a necessity. The underlying motivations for this may include the prevention of unauthorized access to private areas, the protection of green spaces from potential damage caused by human or animal activity, and the delineation of designated pathways within the environment. The available evidence suggests that, in most cases, the absence of such barriers could be considered preferable. However, even in situations where their implementation is deemed essential, there is a potential for enhancing the preference and restorative qualities of the environment.

This study demonstrated that when confronted with no alternative but to install a fence, individuals often opt for a fence of reduced height, probably in an effort to mitigate its perceptible presence. This observation implies that when maintaining a fence at a certain level is not imperative, such as when there is no intent to impose stringent restrictions on access, it is more advantageous to employ the fence at its minimum possible height. For instance, when delineating passing routes, the implementation of a diminutive fence that does not impede the view of the greenery may suffice.

Our findings demonstrate how specific streetscape vegetation design approaches engage ART's restorative mechanisms in ways that may yield both immediate and sustained psychological benefits. The strong preference for trees, primarily driven by fascination, aligns with ART's proposition that environments sustaining gentle, effortless attention can facilitate repeated restorative experiences [8]. This suggests urban tree plantings along daily routes could provide cumulative benefits by regularly interrupting cognitive fatigue. Similarly, the optimal medium diversity arrangements offer the theorized balance of coherence (reducing cognitive load through legibility) and fascination (maintaining interest through complexity) that ART posits as crucial for long-term restoration potential.

## 4.5 VR and Computer-generated environment

In contrast to numerous studies that utilize panoramic images or 360-degree videos to assess environmental influences, our study benefited from a 3D computer-generated environment. A salient benefit of computer-generated VR in this study was its capacity to generate and compare environmental conditions that are logistically unfeasible or exceedingly difficult to reproduce in 360-degree media or photomontages. For instance, our design necessitated direct comparisons of identical streetscapes with systematically varied vegetation (e.g., trees vs. bushes vs. grass) under identical lighting, weather, and spatial conditions. Achieving this objective with 360-degree imagery would necessitate identifying real-world locations that differ solely in vegetation type. However, this is a highly challenging constraint, given the inherent natural variability in urban environments. Despite the utilization of photomontage techniques, preserving visual coherence (e.g., shadows, perspective, scale) across comparisons would inevitably introduce artifacts that could potentially confound participant responses. The utilization of 3D modeling ensured the meticulous control and comparability of all conditions, thereby isolating vegetation as the sole variable of interest.

Furthermore, the implementation of a 3D environment offers a distinct advantage, enabling users to navigate and observe the environment from diverse vantage points, a capability not feasible in experiments which rely on panoramic images or 360-degree videos. While our study allowed physical walking within the tracked space, the complete synchronization between participant motion and visual feedback (without any autonomous scene movement) minimized cybersickness risks. This was confirmed by zero discomfort reports among participants. The walking interaction enhanced ecological validity while maintaining the experimental control advantages of computer-generated VR over passive 360° media.

Although the controlled virtual approach effectively isolates the visual design factors central to this study, real-world settings introduce additional sensory dimensions such as tactile interaction with vegetation, olfactory stimuli, and microclimatic variation, that may modulate restorative experiences. As demonstrated in previous validation studies, strong correlations have been identified between VR and real-world preference rankings for comparable vegetation designs [63,86]. These findings suggest that our visual design principles provide a robust foundation for implementation. However, subsequent research endeavors could meaningfully build upon this foundation by means of the following extensions: first,

to examine how our specific vegetation configurations perform in physical settings, and second, to investigate how non-visual sensory cues interact with the visual design principles identified in this study.

### 4.6  Limitations

It is imperative to acknowledge the limitations inherent in the study's design when interpreting the findings. While the virtual reality environment controlled key variables, it necessarily simplified biological complexity (e.g., seasonal changes, plant growth dynamics) present in real-world settings. Additionally, the evaluation of participants' sense of presence using standardized questionnaires was not conducted, which could have provided further ecological validity.

It is important to acknowledge that, although the participant sample size aligns with typical norms for VR studies, there might be an inadequate representation of demographic diversity. Additionally, while the prior experience of VR had the potential to influence responses, the sample size of VR-experienced participants (n*= 14) was insufficient for conducting meaningful subgroup comparisons. While a visual assessment of the results suggests a negligible effect on the outcomes, it is recommended that further research investigate how varying levels of VR experience impact behavioral outcomes.

Another limitation of this study concerns the interpretation of the ART-based statements. While the items were adapted directly from the validated Perceived Restorativeness Scale and are intended to be accessible to non-specialists, no additional explanations or comprehension checks were provided to participants. As a result, some variation in how participants understood the four ART dimensions cannot be ruled out. Future studies may benefit from including short examples or pre-task comprehension checks to ensure more consistent interpretation.

With regard to the fence effect, while the present study elucidates the manner in which this element influences restorative potential, real-world applications may necessitate the balancing of these findings against contextual needs, such as security or privacy. This constitutes a fundamental direction for subsequent field studies. Concerning the vegetation type, while this study has shown the trees' consistent preference across various arrangement and diversity conditions, further research could examine how specific arboreal attributes (e.g., height, canopy size, and seasonal variation) or other contextual differences modify these effects.

### 5.  Conclusion

The present study employed a choice experiment in a 3D virtual environment to evaluate preference for different characteristics of streetscape vegetation, including fences, vegetation type, diversity, and arrangement. The findings indicated that the material, height, and presence of greenery can influence the appreciation of fences. Nevertheless, the mere presence of a fence, even a low one, constructed of wood and adorned with vegetation, had a detrimental effect on the fascination exerted by the vegetation and the feeling of being attuned to the scene. Additionally, the study illuminated the impact of vegetation type, plant diversity and arrangement in small urban green spaces. The strong preference for trees observed in this study should be considered an important aspect when designing these spaces. It is also recommended that designers and professionals consider that very high or very low levels of diversity and regularity of arrangement are less favorable than an intermediate level. The use of 3D environments and immersive virtual reality in the present study demonstrated the potential for these tools to be employed in choice experiments and environmental preference studies. Further studies could be conducted with similar methods in other urban contexts to evaluate other characteristics, such as tree species, color, size, density and more.

### Supporting information

**S1 Data.  Raw data choice task Fence.**
(XLSX)

**S2 Data.  Raw data choice task Vegetation.**
(XLSX)

## Author contributions

**Conceptualization:** Pooria Baniadam, Ignacio Requena-Ruiz, Jean-Marie Normand, Daniel Siret, Franck Mars.

**Data curation:** Pooria Baniadam.

**Formal analysis:** Pooria Baniadam, Franck Mars.

**Funding acquisition:** Pooria Baniadam, Daniel Siret, Franck Mars.

**Investigation:** Pooria Baniadam, Ignacio Requena-Ruiz, Jean-Marie Normand, Daniel Siret, Franck Mars.

**Methodology:** Pooria Baniadam, Ignacio Requena-Ruiz, Jean-Marie Normand, Daniel Siret, Franck Mars.

**Project administration:** Pooria Baniadam, Daniel Siret, Franck Mars.

**Resources:** Pooria Baniadam, Ignacio Requena-Ruiz, Jean-Marie Normand, Daniel Siret, Franck Mars.

**Software:** Pooria Baniadam.

**Supervision:** Ignacio Requena-Ruiz, Jean-Marie Normand, Daniel Siret, Franck Mars.

**Validation:** Pooria Baniadam, Ignacio Requena-Ruiz, Jean-Marie Normand, Daniel Siret, Franck Mars.

**Visualization:** Pooria Baniadam.

**Writing – original draft:** Pooria Baniadam.

**Writing – review & editing:** Pooria Baniadam, Ignacio Requena-Ruiz, Jean-Marie Normand, Daniel Siret, Franck Mars.

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
