## [Decision Letter · Decision Letter 0]

23 Mar 2025

PONE-D-25-01612Using virtual reality to study preference and restorativeness for streetscape vegetation designPLOS ONE

Dear Dr. Baniadam,

Thank you for submitting your manuscript to PLOS ONE. After careful consideration, we feel that it has merit but does not fully meet PLOS ONE’s publication criteria as it currently stands. Therefore, we invite you to submit a revised version of the manuscript that addresses the points raised during the review process.

We look forward to receiving your revised manuscript.

Kind regards,

Tianheng Shu, PhD

Academic Editor

PLOS ONE

Journal Requirements:

Additional Editor Comments:

The reviewers generally agree that you have explored an intriguing topic and the overall quality of your research is high. However, they have also raised some concerns that need to be addressed, such as clarifying the methodology and discussing additional factors that may have influenced fence preferences. Please carefully address each point.

Reviewers' comments:

Reviewer's Responses to Questions

**Comments to the Author**

1. Is the manuscript technically sound, and do the data support the conclusions?

Reviewer #1: Yes

Reviewer #2: Yes

Reviewer #3: Yes

Reviewer #4: Yes

2. Has the statistical analysis been performed appropriately and rigorously? 

Reviewer #1: Yes

Reviewer #2: Yes

Reviewer #3: No

Reviewer #4: Yes

3. Have the authors made all data underlying the findings in their manuscript fully available?

Reviewer #1: Yes

Reviewer #2: Yes

Reviewer #3: Yes

Reviewer #4: Yes

4. Is the manuscript presented in an intelligible fashion and written in standard English?

Reviewer #1: Yes

Reviewer #2: Yes

Reviewer #3: No

Reviewer #4: Yes

5. Review Comments to the Author

Reviewer #1: My review will primarily focus on the VR methodology employed in this study, as it aligns with my field of research and expertise. While the manuscript presents a well-structured investigation into streetscape vegetation preferences and restorativeness, my assessment will concentrate on the validity, strengths, and potential areas for improvement in the use of VR as a research tool within this context.

I have identified the following three key strengths of the VR approach used in this study:

• Use of immersive VR: The study correctly identifies that VR enhances realism and presence, offering a significant advantage over traditional 2D imagery or photo-montage methods. The choice of the HTC Vive Pro for high-quality rendering is appropriate and ensures a reasonable level of immersion.

• Experimental control: VR allows for controlled manipulation of variables (e.g., fence attributes, vegetation type, arrangement) while keeping all other environmental factors constant. The method of allowing participants to switch between different conditions in real-time adds a comparative dimension that strengthens internal validity.

• Integration of a VR-based questionnaire: Embedding the questionnaire within the VR environment helps maintain immersion and reduces the potential bias associated with switching between VR and external response modes.

The following are recommendations and suggestions for Improvement:

• Lack of details on scene realism and presence: The study references presence but does not provide any quantitative assessment of it (e.g., using a standardized presence questionnaire). Since VR studies rely on the feeling of "being there," assessing presence is crucial. It is also unclear whether the virtual environments were designed to match photorealistic standards or were more stylized. The level of realism or abstraction can impact perceived restorativeness. I recommend including a validated measure of presence (e.g., the Slater-Usoh-Steed questionnaire) and providing details on the graphical fidelity of the virtual scenes.

• Limited justification for using 3D-modeled VR over 360-degree images or videos: While the study briefly mentions the advantages of computer-generated VR over 360-degree media, it does not explicitly justify why this approach was necessary for studying streetscape vegetation. Previous research has shown that 360-degree real-world imagery can be equally effective in studying environmental preferences while maintaining higher ecological validity. I suggest providing a more detailed discussion on why 3D modeling was chosen over other VR methods and its impact on ecological validity.

• VR-related limitations and interaction: The study does not address whether participants experienced discomfort, simulator sickness, or hardware limitations (e.g., field of view restrictions, rendering lag). I recommend including a discussion on potential VR-related limitations, such as motion sickness or interaction constraints, and how they may have affected results.

• Participant familiarity with VR: While the manuscript acknowledges participants' varying levels of VR experience, it does not analyze whether prior familiarity influenced their choices. Participants with extensive VR experience might have a different level of comfort and engagement with the virtual scenes compared to novices. I encourage the authors to include metrics that assess whether VR experience had a significant effect on participant responses and consider it as a potential confounding factor.

Reviewer #2: This study investigates the effects of street vegetation design on preference and restorative perception using virtual reality (VR) technology, a topic with practical significance.The article has a clear structure, substantial workload, relatively rigorous experimental design, appropriate data analysis methods, and provides valuable reference for urban landscape design. However, there are still some aspects worth further optimization to enhance the scientific nature and persuasiveness of the research.

1. At the end of the introduction, after stating the research purpose, I believe it is necessary to present the research questions and emphasize the innovation and significance of the study.

2. Furthermore, in the methodology section, it is suggested to include a research framework diagram to help readers better understand the study.

3. In the participants section, it is recommended to add the rationale for sample effect size selection.

4. In section 2.4, it is suggested to add scenario images of subjects during the experiment to enhance the readability of the article.

5. The author thoroughly discusses the results in the discussion section; however, the limitations of the research are not mentioned. It is suggested to supplement this content.

6. It is suggested that the author further explore the theoretical value of the research in the discussion section, as well as some specific practical values for urban landscape design in reality.

Reviewer #3: This manuscript based on Attention Restoration Theory, employs VR technology to explore the factors influencing individuals' preferences for the design of streetscape vegetation. It demonstrates a solid theoretical foundation and appropriate data processing. However, several suggestions and concerns should be addressed:

1. The title of the manuscript specifies "streetscape vegetation design", yet the scope of the study appears to extend beyond vegetation alone. It is recommended reconsider the title to more accurately reflect the study’s focus.

2. The current Introduction should be reorganized. It is suggested that the literature review be presented under a separate heading, while the definitions of key terms may be incorporated within either Introduction or Method section. Additionally, a research framework describing experiment design, data processing, and statistical method can further clarify the methods in this manuscript.

3. Does the experiment exclusively rely on the selection of reasons from the four predefined options? What are the participants' academic backgrounds? Were adequate explanations provided to ensure participants fully understood the meaning of each option?

4. The presentation of Results requires refinement. At present, a substantial number of figures are used, yet the analytical approach appears similar across them. It is recommended that the figures and tables be consolidated to increase the information density of each one. Additionally, the result analysis should strengthen the connection between the results and the theoretical foundation.

5. In particular, it is suggested that the authors explicitly articulate the theoretical and practical contributions of the findings for each research aspect (i.e., fence, vegetation type, plant diversity and arrangement).

6. The discussion and conclusion do not address potential differences between embodied experience and virtual experience. Considering the use of VR technology in the study, it is recommended that the authors incorporate a discussion on this aspect.

7. The numbering of references, figures, and their corresponding textual citations should be double checked.

Reviewer #4: The study explores the usage of VR in the context of studying preference and restorativeness for streetscape vegetation design. Overall, the article is interesting and well written.

Here are some of my questions to the authors:

• Did the authors explore additional factors that may have influenced fence preferences, such as security, aesthetics, or privacy concerns? Although the “absence of a fence was the most preferred option,” preferences can shift depending on environmental and social contexts. Was any analysis conducted to assess how these factors influenced participant choices?

• Regarding tree preferences, did the authors explore specific attributes of trees (e.g., height, movement, shade) that contributed to this preference? Also, could the authors explain how tree preferences could vary between rural and non-rural areas? Lastly, was this preference affected by whether the vegetation arrangement appeared natural or was intentionally designed?

• I found it interesting that the authors gave such a detailed description of the fences, like: “The wooden fences were made of pickets of equal width (approximately 10 cm) with gaps between them and connected by rails.”. Were these dimensions a specific design choice by the authors, or were they based on legal architectural regulations or industry standards? Also, did all the design choices in this study follow existing regulations, or were they just selected for research purposes?

• The study lacks a clear connection to Attention Restoration Theory (ART). From my understanding, ART is often used as a framework for designing green spaces in urban areas to enhance mental well-being. However, the authors do not explore how their findings might contribute to long-term cognitive and psychological benefits for the population.

Discussing the sustained impact of different vegetation types and arrangements on attention restoration would strengthen the study's relevance to ART.

• Adding a Limitation section would be helpful for readers and researchers to understand the challenges faced in the study and learn from the authors' experiences.

6. PLOS authors have the option to publish the peer review history of their article (what does this mean? ). If published, this will include your full peer review and any attached files.

**Do you want your identity to be public for this peer review?** For information about this choice, including consent withdrawal, please see our Privacy Policy .

Reviewer #1: No

Reviewer #2: No

Reviewer #3: No

Reviewer #4: No

---

## [Author Response · Author response to Decision Letter 1]

7 Jun 2025

The response letter to the reviewer is submitted as a PDF file.

---

## [Decision Letter · Decision Letter 1]

9 Jul 2025

PONE-D-25-01612R1Using virtual reality to study preference and restorativeness for streetscape vegetation designPLOS ONE

Dear Dr. Baniadam,

Thank you for submitting your manuscript to PLOS ONE. After careful consideration, we feel that it has merit but does not fully meet PLOS ONE’s publication criteria as it currently stands. Therefore, we invite you to submit a revised version of the manuscript that addresses the points raised during the review process.

We look forward to receiving your revised manuscript.

Kind regards,

Tianheng Shu, PhD

Academic Editor

PLOS ONE

**Journal Requirements:**

Reviewers' comments:

Reviewer's Responses to Questions

**Comments to the Author**

1. If the authors have adequately addressed your comments raised in a previous round of review and you feel that this manuscript is now acceptable for publication, you may indicate that here to bypass the “Comments to the Author” section, enter your conflict of interest statement in the “Confidential to Editor” section, and submit your "Accept" recommendation.

Reviewer #1: All comments have been addressed

Reviewer #2: All comments have been addressed

Reviewer #3: (No Response)

2. Is the manuscript technically sound, and do the data support the conclusions?

Reviewer #1: Yes

Reviewer #2: Yes

Reviewer #3: Yes

3. Has the statistical analysis been performed appropriately and rigorously? 

Reviewer #1: Yes

Reviewer #2: Yes

Reviewer #3: Yes

4. Have the authors made all data underlying the findings in their manuscript fully available?

Reviewer #1: Yes

Reviewer #2: Yes

Reviewer #3: Yes

5. Is the manuscript presented in an intelligible fashion and written in standard English?

Reviewer #1: Yes

Reviewer #2: Yes

Reviewer #3: Yes

6. Review Comments to the Author

**Reviewer #1:**  I appreciate the authors’ thoughtful and substantial revisions in response to the previous feedback. The revised manuscript is significantly improved, with greater clarity in research aims, methodological rigor, and justification for the use of immersive VR. Specifically, I commend the authors for:

- Providing a much clearer justification for employing 3D-modeled VR over 360° imagery, particularly in relation to experimental control and ecological validity.

- Expanding the discussion of VR-related limitations and confirming that no participant discomfort or simulator sickness was reported.

- Including a transparent acknowledgment of the absence of a standardized presence measure and the role of prior VR experience, along with a reasonable explanation for why further analysis was not conducted due to sample size constraints.

While the manuscript still does not incorporate a formal assessment of presence or detailed discussion of visual realism (e.g., photorealism vs. stylization), these limitations are now clearly acknowledged and do not detract from the overall validity of the findings.

I find the authors’ responses to be satisfactory and support the publication of the manuscript in its current or near-final form.

**Reviewer #2:**  (No Response)

**Reviewer #3:**  Thank you for the responses and the substantial improvements made to the manuscript. I believe the paper has significantly improved. However, I would like to raise the following two remaining points that should be addressed in the revision:

1. While the authors provided basic demographic information, it remains unclear what academic or disciplinary backgrounds the participants had. This is particularly relevant given the use of theoretical constructs such as the ART dimensions. Were participants from relevant fields? Were any measures used to ensure participants fully understood the meaning of the four ART-based statements?

2. The standalone section on “Key Terms” feels disjointed from the overall narrative. I suggest integrating these definitions directly into the literature review, where the terms can be introduced in context.

7. PLOS authors have the option to publish the peer review history of their article (what does this mean? ). If published, this will include your full peer review and any attached files.

**Do you want your identity to be public for this peer review?** For information about this choice, including consent withdrawal, please see our Privacy Policy .

Reviewer #1: No

Reviewer #2: No

Reviewer #3: No

---

## [Author Response · Author response to Decision Letter 2]

23 Aug 2025

The response to the reviewers has been uploaded as a PDF file.

---

## [Editor Report · Decision Letter 2]

1 Sep 2025

Using virtual reality to study preference and restorativeness for streetscape vegetation design

PONE-D-25-01612R2

Dear Dr. Baniadam,

We’re pleased to inform you that your manuscript has been judged scientifically suitable for publication and will be formally accepted for publication once it meets all outstanding technical requirements.

Kind regards,

Tianheng Shu, PhD

Academic Editor

PLOS ONE

Additional Editor Comments (optional):

Reviewers' comments:

---

## [Editor Report · Acceptance letter]

PONE-D-25-01612R2

PLOS ONE

Dear Dr. Baniadam,

I'm pleased to inform you that your manuscript has been deemed suitable for publication in PLOS ONE. Congratulations! Your manuscript is now being handed over to our production team.

Kind regards,

on behalf of

Dr. Tianheng Shu

Academic Editor

PLOS ONE